# Biorefinery of Biomass of Agro-Industrial Banana Waste to Obtain High-Value Biopolymers

**DOI:** 10.3390/molecules25173829

**Published:** 2020-08-23

**Authors:** Carlos Redondo-Gómez, Maricruz Rodríguez Quesada, Silvia Vallejo Astúa, José Pablo Murillo Zamora, Mary Lopretti, José Roberto Vega-Baudrit

**Affiliations:** 1National Laboratory of Nanotechnology LANOTEC, 1174-1200 Pavas, San José, Costa Rica; redox06@gmail.com; 2School of Chemistry, National University of Costa Rica (UNA), 86-3000 Heredia, Costa Rica; marirquesada@gmail.com (M.R.Q.); vallejo.sil.15@gmail.com (S.V.A.); josepablomu@gmail.com (J.P.M.Z.); 3Laboratorio de Técnicas Nucleares Aplicadas a Bioquímica y Biotecnología, Centro de Investigaciones Nucleares-Facultad de Ciencias, UDELAR University, cp1140 Montevideo, Uruguay; mlopretti@gmail.com

**Keywords:** biorefinery, residue, agro-industry, high-value products, banana

## Abstract

On a worldwide scale, food demand is increasing as a consequence of global population growth. This makes companies push their food supply chains’ limits with a consequent increase in generation of large amounts of untreated waste that are considered of no value to them. Biorefinery technologies offer a suitable alternative for obtaining high-value products by using unconventional raw materials, such as agro-industrial waste. Currently, most biorefineries aim to take advantage of specific residues (by either chemical, biotechnological, or physical treatments) provided by agro-industry in order to develop high-value products for either in-house use or for sale purposes. This article reviews the currently explored possibilities to apply biorefinery-known processes to banana agro-industrial waste in order to generate high-value products out of this residual biomass source. Firstly, the Central and Latin American context regarding biomass and banana residues is presented, followed by advantages of using banana residues as raw materials for the production of distinct biofuels, nanocellulose fibers, different bioplastics, and other high-value products Lastly, additional uses of banana biomass residues are presented, including energy generation and water treatment.

## 1. Introduction

The rising development of industries all over the world has brought a consequential increase in their residue generation, especially in the field of agro-industry. This waste can be denominated as “food supply chain waste” (FSCW) and can be defined as “organic material produced for human consumption lost or degraded primarily at the manufacturing and retail stages” [1]. This concept has emerged in the context of the current vast inefficiency of the food supply chain business. For instance, the Food and Agriculture Organization (FAO) revealed in 2011 that up to a third of the food aimed at human consumption is wasted every year globally [2]. The environmental and economic impacts of this worrying situation have driven the development of technologies pursuing not only conventional waste management and disposal, but also the extraction of as much value as possible out of any given agro-industrial waste. 

### 1.1. Agro-Industry Residues as Biomass Sources

Agro-industries have slowly come to realize that valorization of biomass residues (either by using them as raw materials for the development of high-value products, or investing in recirculating processes that make use of these residues to obtain income in the long run) is not only beneficial from an environmental perspective, but can also help to minimize economic losses or even raise the net value of companies.

Inadequate treatment of these biomass residues has a negative impact on the environment, mainly generating greenhouse gases, contaminating water sources, and causing ecological problems [3,4]. Biorefinery technologies rise as suitable alternatives to mitigate these impacts as they can assist on reducing waste volume [5], but also on producing high-value goods out of revalorized biomass waste for circular economy purposes.

Waste valorization converts polymeric substrates into useful products such as chemicals, materials, and fuels, often by extraction, chemical conversion, or degradation. Historically, the utilization of complex biomasses led the way in pulp and paper production, or biotechnological production of furfural, ethanol, or short chain organic acids since the 19th century [6]. However, it was not until the 1990s when the term “biorefinery” became widespread in the industry, once biomass started to be used as a source of higher-value products [7,8].

### 1.2. The Concept of Biorefinery

Out of the many published definitions for the term “biorefinery”, perhaps the one from the American National Renewable Energy Laboratory (NREL) seems to fit best the approach of this review: “A biorefinery is a facility that integrates biomass conversion processes and equipment to produce fuels, power, and chemicals from biomass” [6]. Definitions such as this one comprise the conversion of biomasses not only into biofuels, biopolymers, high-value products, and fine chemicals, but also include the generation of power (heat and electricity) analogous to today’s petroleum-based refineries [7].

In addition to lignocellulosic biomass-based industries, more sectors have shown interest in applying biorefinery approaches to organic waste, for instance: food waste [9], nonedible oils [10] sewage sludge [11], and municipal solid waste [12] just to name a few. Biorefinery technologies enable more efficient use of agricultural resources and sustainable food production [9]. Taking advantage of biorefinery technologies represents a valuable strategy for agro-industries and governments; hence, they can easily navigate the challenges of the green economy era. 

## 2. Applications of Agro-Industrial Residues

Agro-industrial waste is mainly composed of lignocellulose biomass. Lignocellulose waste has been gaining increasing attention due to its mechanical and thermal properties, renewability, wide availability, non-toxicity, low cost, and biodegradability [13,14]. The vast range of lignans and celluloses comprised in agro-industrial residues grants them tremendous potential as feedstocks for chemical and biotechnological conversion processes. For instance, enzymatic breakdown of cellulose and hemicelluloses into glucose and xylose allows further fermentation of these monosaccharides into ethanol by fermentative microorganisms. Furthermore, pyrolysis and anaerobic digestion of lignocellulose biomasses can yield combustion gases such as H_2_ and CH_4_ [13,15,16].

Currently, organic and agro-industrial residues take up a large portion of overall global waste (Figure 1a), which is one of the reasons to make good use of it. The abundance of biomass feedstocks gives a positive prospect for their future utilization in biorefinery technologies [17]. Estimates on biomass crop residue flows in Latin America show that most of lignocellulose containing biomasses are mostly made of maize, soybean, and sugarcane residues [18]; banana residues are not found within the main agro-industrial residues of developing countries to be used as biorefinery biomass sources (Figure 1b), though in many locations banana waste treatment remains a problem that needs to be addressed [19], as we discuss in the following section.

### 2.1. Generation of Banana Residues

A banana plant is a tall and sturdy herbaceous plant, with a succulent and very juicy tubular stem, composed of leaf-petiole sheaths consisting of long and strongly overlapping fibers called pseudostem. Each pseudostem bears fruit only once, before dying and being replaced by a new one; this pseudostem consists of concentric layers of a leaf sheath and a crown of large leaves [15]. 

Banana biomass mainly consists of four elements, namely: pseudostems, leaves, rachis, and skins. Additionally, a significant number of rejected bananas provide starchy feedstock to feed biorefineries. Feedstock derived from rejected bananas can reach up to 30 wt% of the total production (remaining unsold overripe fruits also fall in this category) [15]. All these biomass residues are normally dumped in rivers, oceans, landfills, and unregulated dumping grounds, creating huge decaying deposits that can lead to the spread of diseases, contamination of water sources, generation of foul odors, and attraction of rodents, insects, and scavengers. Some of the possible ways that enable the utilization of banana waste include compost production and food wrapping. However, these solutions do not always prevent the material from reaching the wasteland after serving its purpose. Recently, a craft type paper of good strength has been made from crushed, washed, and dried banana pseudostems [15].

### 2.2. Banana Residues in Central and Latin America

Banana is one of the most cultivated fruit crops worldwide (~106.7 million tons of production in 2013). Many industries take advantage of banana pulp, but discard lignocellulosic biomass, including pseudostems, stalks, leaves (normally found in the field), and rachis of the fruit bunches (gathered usually in the packing plants). Leaves, the pseudostem, stalk, and peel generate a huge amount of waste [20]; for instance, banana peels account for more than 41.3 million tons per year, therefore serving as a potential biomass feedstock [21].

The estimated amount of agricultural residue available in Central America in 2011 was about 192 Petajoules (PJ); the countries with the highest energy potentials are Guatemala with 79 PJ and Honduras with 29 PJ, followed by Costa Rica with 22 PJ. Banana residues represent an important fraction of these wastes, as the Central American region provides excellent environmental conditions for optimal development for the banana plant. In this fashion, banana crops rank in the top six agriculture residues in countries such as Belize, Costa Rica, Guatemala, Honduras, and Panama [21].

Nevertheless, Central American producers are far more focused on commercializing the crop itself than valorizing the corresponding waste. In 2011, about 2.9 million tons of banana residue (wet basis) were produced in Central America [21]. However, it is worth mentioning that these residues cannot be fully recovered, as part of them must be left in situ to avoid soil degradation (i.e., reduction of carbon stock in the soil), while other residues have found uses as fertilizers, fodder, and domestic fuel [21]. Nonetheless, there is still a large proportion that can find applications as biorefinery feedstocks. 

Other Latin American nations face similar realities when it comes to banana production (Figure 1b). For instance, Brazil produces around 82.8 million tons of bananas annually; each produced ton leaves behind 100 kg of rejected fruit and some 4 tons of lignocellulosic waste (3 tons pseudostems, 480 kg leaves, 160 kg rachis, and 440 kg skins) [22]. 

Nations such as Ecuador have come up with a series of initiatives regarding bioethanol production using lignocellulosic biomass from banana crops. For instance, Guerrero and collaborators developed a process of production of bioethanol from banana rachis with a positive energy balance [23,24]. Figure 2 presents schematics of the production and use of second-generation ethanol from banana waste and its further blending with regular gasoline, this study employed a Well-to-Wheel (WtW) perspective and concludes that this strategy has great potential to reduce greenhouse gas emissions and fossil depletion, as a consequence of an overall positive energy balance for the process [23]. Bioethanol production from banana wastes is further discussed in Section 3.1 of this review.

### 2.3. Potential Biorefinery Use of Different Banana Residues

Before addressing the possible ways to convert banana residues into high-value products through biorefinery, it is important to describe their physicochemical properties, as this information allows for their maximum exploitation as raw materials and would help in developing more eco-friendly approaches too. Banana peel and rachis waste are composed mainly of biopolymers such as lignin, pectin, cellulose, hemicellulose [25], fiber, proteins, and some low-molecular-weight compounds such as chlorophylls, phenolic compounds, water-soluble sugars, and minerals. 

Table 1 shows the composition of banana peel and rachis on a dry matter basis. It is worth mentioning that the high moisture content of banana residues promotes their biodegradability before processing, thus affecting their handling, transportation, storage, and further uses in biorefinery technologies [20]. The composition of fruit peel residues varies according to species, seasonal variations, geographic location, variety, and stage of maturation [20]. Lignin contents are greater in banana rachis [23], and banana leaves are rich in holocellulose, hemicellulose, and lignin, all promising compounds for biorefinery processing. Lignin is particularly valuable, accounting for 25 wt% of banana leaves, which is higher than other important agro-industry residues such as cotton or straw [26]. Banana rachis and pseudostem residues can be used as biomass feedstocks, both biomasses have a high content of carbohydrates such as hemicellulose, starch [23], and lignin [19]. Banana stem residues also contain lignins, glucans, and most abundantly xylans and ashes [27]. 

Though mechanical [28] and hydrothermal pretreatments of these residues are still energy and time consuming [13], they have proved necessary for further steps in biorefinery operations. For instance, steam explosion pretreatment increases cellulose content compared to raw materials (from 20.1 to 54.4 wt% going from raw to pretreated pseudostem, and 26.1 to 57.1 wt% going from raw to pretreated rachis); pretreatment also increases free glucose content for further biotransformations [29].

## 3. High-Value Products Obtained from Banana Residues

As shown above, a large number of parts of banana residues can be used as biomass sources; chemical composition of the raw material in question will determine its further suitability as a biomass source. In this section we describe a series of high-value products obtained from banana residues via biorefinery technologies.

### 3.1. Biofuels 

Chemical composition of banana stems provides an indicator of their feasibility for production of fermentable sugars as a function of moisture content, as well as cellulose and hemicellulose contents. In fact, saccharification and further fermentation of banana lignocellulosic content for ethanol production has been extensively investigated [28,29,30]. Research by Duque and collaborators shows that the potential of ethanol production is 0.259 kg of ethanol per kg of banana stem raw material [31]. Research by Guerrero and co-workers has found high solid loading, low enzyme dosage and a short period conversion process as optimal conditions for bioethanol production from banana pseudostem and rachis, yielding ethanol solutions of 4.0 *v*/*v* % (87% yield) and 4.8 *v*/*v* % (74% yield), respectively [29]. 

Another study by Ingale and co-workers also employed banana stem waste and found that alkali treated banana pseudostems followed by enzymatic saccharification yield higher contents of reducing sugars than those alkali treated only, the corresponding increase in ethanol production was observed (Figure 3) [32]. Not only has lignocellulosic waste from banana production been transformed into ethanol, but fermentation of banana pulp and fruit can yield comparable biofuel efficiencies as corn [28].

Even though further technological improvements are still required, bioethanol yield from banana waste presents a promising alternative for the production of this biofuel. Further research on improved enzymatic cocktail formulations, more robust microorganism strains, as well as optimization of industrial conditions, such as reaction time, water content, and ethanol separation technologies [30], are still required [28].

### 3.2. Fibers for Mechanical Reinforcement

Banana fiber has traditionally found a place in a number of manufactured products such as paper, ropes, table mats, and handbags [15]. Though these materials are inexpensive, biodegradable, and produced from renewable sources, biorefinery approaches offer the possibility to generate higher value outcomes from their raw material. In this fashion, lignocellulosic micro/nanofibers (LCMNF) can be produced from banana leaf residues; Tarrés and co-workers’ results show that banana leaves can yield up to 82.44% LCMNF, a significantly high value compared to other agricultural wastes that typically yield around 15% [26].

LCMNFs from banana leaf residue have been used to restore mechanical properties of recycled fluting paper. A study found that incorporation of only 1.5 wt% of LCMNF can restore the original properties of fluting paper, with a low impact in pulp drainability, while increasing the life span of the resulting recycled products [33].

### 3.3. Nanocellulose Fibers

Banana residues are a great source of cellulosic materials (see Table 1); cellulose provides stiffness and strength to the plants’ structure, and approximately a third of the plant’s anatomy is composed of this polysaccharide [34,35,36,37]. Since cellulose is greatly present in banana peel and rachis (Table 1), those represent biomass sources suitable to obtain nanocellulose fibers (NCFs).

NCFs exhibit many attractive physicochemical properties such as a high bending strength (~10 GPa), a Young’s modulus of approximately 150 GPa, a high aspect ratio, and a high specific surface area. Therefore, NCFs have been used as reinforcements for polymer matrixes [38], and as additives for papermaking. Suspensions of NCFs improve the mechanical strength and density of paper while reducing its porosity [15]. The surface of NCFs is decorated with polar hydroxyl groups, which confer high moisture adsorption capacity and surface reactivity [39]. NCFs present a strong potential as oil-water suspension stabilizers in the food industry [35]; as mechanical reinforcement [40] in drug delivery, enzyme supports, biosensors [41], and scaffolds for tissue engineering applications [39,40,42].

Acid hydrolysis of cellulose is the most common process for obtaining NCFs [43], as fractions containing amorphous material can be hydrolyzed with HCl and sulfuric acid, while those containing crystalline cellulose are typically recovered by centrifugation [44,45]. NCFs have been isolated from banana peel using different processes, involving alkaline treatment and bleaching, followed by acid hydrolysis with sulfuric acid and high-pressure homogenization [46]. Transmission electron microscopy (TEM) and atomic force microscopy (AFM) investigations have revealed the clearance of large amounts of amorphous materials to afford highly crystalline NCFs, these NCFs showed good cytocompatibility with human epithelial colorectal adenocarcinoma cells (Caco-2 cell line) in concentrations below 1000 mg/mL, these NCFs exhibit promising features as reinforcement material in composites too (Figure 4) [46].

### 3.4. Bioplastics

Poly-hydroxybutyrates (PHBs) are value-added biocompatible, biodegradable, thermoplastic biopolymers that can be synthesized by microorganisms from diverse carbon sources. Polysaccharides in banana peel can be either chemically or microbiologically transformed into PHBs [8], these biopolymers are hydrophobic, and bear similar mechanical properties to polypropylene or polyethylene. Getachew and collaborators have reported on a series of strains of *Bacillus* sp. able to yield up to 27 *w*/*w* % of PHB content after the fermentation of hydrolyzed banana peel residues [47]. PHB production from banana waste is still not affordable on an industrial scale, though efforts to couple this production as part of a multiproduct biorefinery are moving the field in this direction; for instance, Naranjo and coworkers reported on how such a kind of integration might save energetic costs and water waste via the fermentation of banana peel hydrolysates using *Burkholderia sacchari* IPT101 [8].

Poly-(*l*-lactic acid) (PLA) is a biodegradable and renewable polyester with many industrial and biomedical applications, including drug delivery systems, bioabsorbable fixation devices, bone regeneration, and tissue engineering scaffolds [48,49,50]. PLA has been obtained through fermentation of banana (and also pineapple) waste hydrolisates using *Lactobacillus casei* (subspecies rhamnosus), and further microwave-assisted polymerization, as reported by Jiménez-Bonilla and collaborators [48]. This direct melt polycondensation method afforded PLA oligomers with low oxidation losses, better stereopurity and lower energetic cost than conventional heating methods (Figure 5) [48].

### 3.5. Enzymes and Food Additives

Banana stalk residues have been valorized in the bio assisted production of enzymes like laccase, different oxidases, and endoglucanases too [13]. For instance, Reddy and co-workers investigated the use of *Pleurotus ostreatus* and *P. sajor-caju*, to produce lignolytic and cellulolytic enzymes such as laccase, lignin peroxidase, xylanase, endo-1,4-β-D-glucanase (CMCase), and exo-1,4-β-D-glucanase using banana wastes as solid substrate fermentation. Both microorganisms originated comparable levels of enzyme activities and patterns of production. 

Leaf biomass was found to be an appropriate substrate (compared to pseudostems) for enzyme production [51]. Banana peel extracts have been studied as antioxidants in fresh orange juices, finding that free radical scavenging capacity increased by adding banana peel extracts to juice formulations. In addition, remarkable increases in antioxidant capacity using 2,2′-azino-bis-(3-ethylbenzothiazoline)-6-sulfonic acid (ABTS) radicals were observed when equal or greater than 5 mg of banana peel extract per ml of freshly squeezed juice was added, though no clear effects were observed in its ability to reduce the extent of lipid peroxidation [52].

## 4. Additional Uses of Banana Residues

### 4.1. Energy Generation

There seems to be a worldwide agreement on shifting towards green energy production and lessening the current dependence on fossil fuels. Agro-industrial biomass sources are a natural choice when it comes to exploring environmentally friendly ways to produce energy, and banana wastes are no exception to this. Banana residues have made it into the energy production sector only in recent times and their potential as energetic biomass is becoming evident; for instance, energy generation from dry banana peel can yield up to 18.89 MJ/kg [20]. The reader is invited to consult reference [19] for an extensive assessment of banana biomass as an energy source in the Central American region.

Currently, there are two approaches for the conversion of banana biomass into energy: thermal and biological conversion [20]. The former involves direct combustion and gasification, while the latter involves anaerobic digestion as shown in Figure 6. Compared to other types of waste substrates (such as human sewage, piggery, or feedlot waste), banana residues produce a very clean form of biogas (mostly made of methane and carbon dioxide, with little noxious odors) [15]. Pisutpaisal and collaborators have demonstrated that size reduction of banana peel raw material and its fungal pretreatment might improve methane yield [16].

Theoretical estimates for potential power generation based on both banana waste and banana peels in Malaysia (by Tock and coworkers for the period 2003–2008) suggest that banana biomass is a suitable renewable energy source in this one and other similar tropical nations. This study calculated potential power assuming that 1 PJ can be converted into 46 MW of electrical energy with 21% electrical conversion efficiency. This study estimates that using whole banana residues might generate some 80–95 MW yearly, while banana peels only would generate 12–25 MW per year (15% and 25% of energy production using the whole residue) as detailed in Table 2 [15].

### 4.2. Water Treatment

Banana peel has been reported to be used as a bio adsorbent for the removal of contaminants such as heavy metals, dyes, and organic pollutants from wastewaters [53]. A study by Pathak and co-workers reports on the adsorptive removal of benzoic acid (BA) and salicylic acid (SA) using banana peel (Figure 7); the authors report on removal efficiencies between 60–80% for the removal efficiency of these contaminants, with the advantage of possible reuse of the banana peel adsorbent in gasification for power generation [54] (though adsorbing contaminants that may be detrimental for biogas quality must be avoided). Banana pith has been used to produce activated carbons to be employed in divalent heavy metal cations and dye removal from aqueous solutions with satisfactory results [55].

## 5. Conclusions

Banana plants exhibit high growth rates and carbon neutrality; therefore, their agro-industrial residues are a promising alternative to be used as feedstock for biorefinery technologies, though challenges regarding composition variations in the wastes must be addressed, i.e., geographic location, plant variety, ripening stage, among others that difficult standardization for biorefinery processing [20]. 

Current technologies require further developments in order to extract as much value as possible from banana waste, for instance, by achieving positive energy balances by full integration of different biorefinery processes. Efforts involving the production of bioethanol and biogas are currently moving in this direction [23,24]. Although efforts are being made on the production of enzymes and food additives derived from different components of banana waste, the most promising potential for these residues rely on high-value biopolymers, i.e., micro and nanofibrillar mechanical reinforcements such as nanocellulose fibers, as well as the biotechnologically and chemically assisted production of bioplastics such as PHBs and PLA. More research on biorefinery approaches will be required if banana residues’ biomass potential is to be increased. As a result of this, small communities from developing countries, as well as agro-industrial, chemical, and pharmaceutical industries, are most likely to benefit.

## Figures and Tables

**Figure 1 molecules-25-03829-f001:**
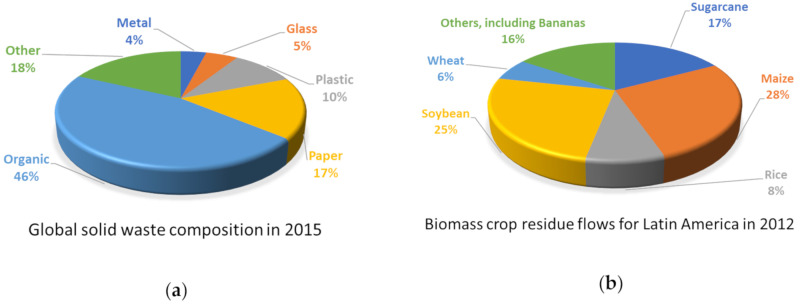
(**a**) Composition of global solid waste in 2015 (adapted from [17]). (**b**) Estimated biomass crop residue flows for Latin America in 2012 (adapted from Table 3.1 in [18]).

**Figure 2 molecules-25-03829-f002:**
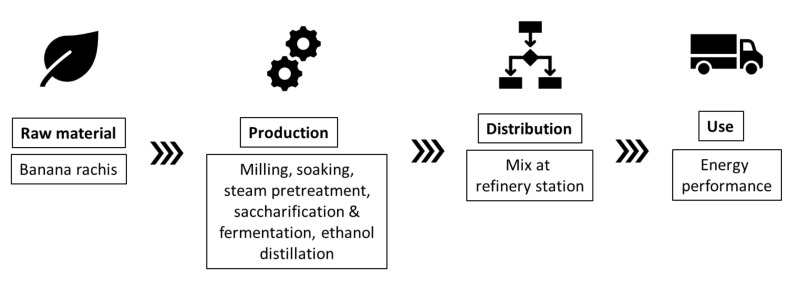
Life cycle system of second-generation ethanol production from banana rachis [23].

**Figure 3 molecules-25-03829-f003:**
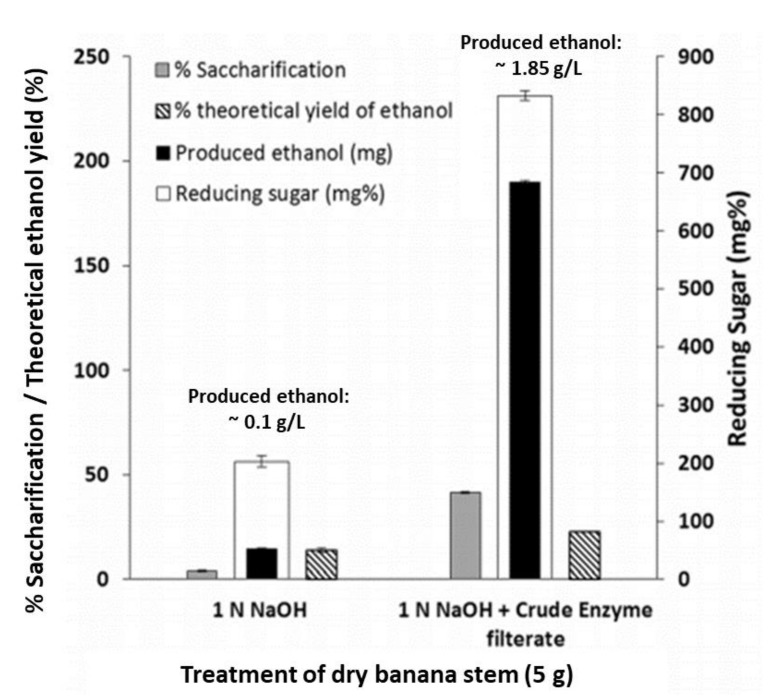
Saccharification percentage and ethanol yield from “alkali treated” and “alkali + enzymatic treatment” banana pseudostems (modified from Reference [32]).

**Figure 4 molecules-25-03829-f004:**
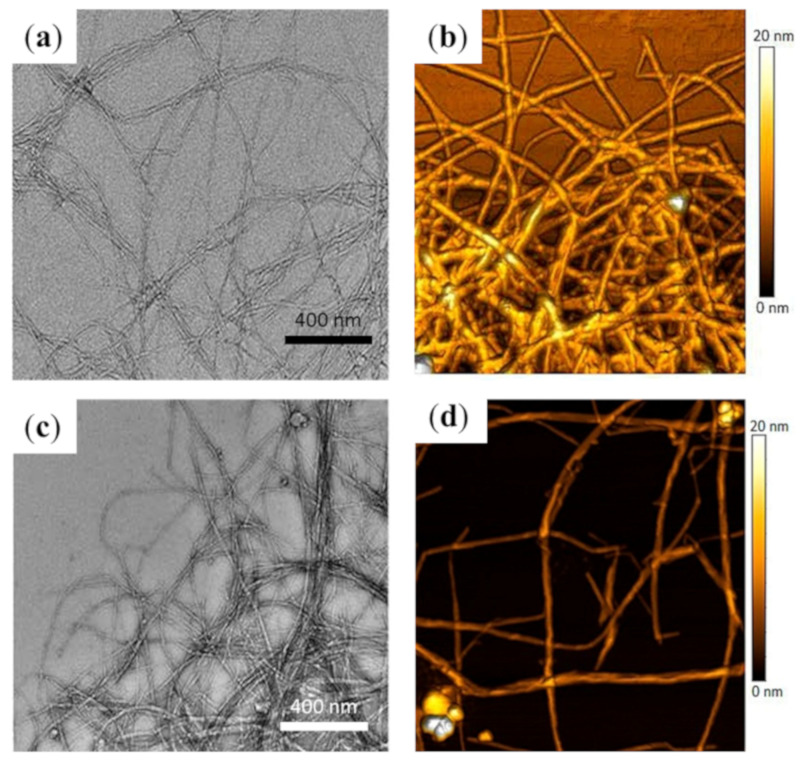
Banana peel-derived nanocellulose fibers (NCFs) produced by chemical hydrolysis without (panels (**a**) and (**b**)) or with mechanical treatment (high-pressure homogenization, panels (**c**) and (**d**)). (**a**) Transmission electron microscopy (TEM) and (**b**) atomic force microscopy (AFM) images of NCFs produced by chemical hydrolysis without mechanical treatment (**c**) TEM and (**d**) AFM images of NCFs produced with mechanical treatment (modified from Reference [46]).

**Figure 5 molecules-25-03829-f005:**
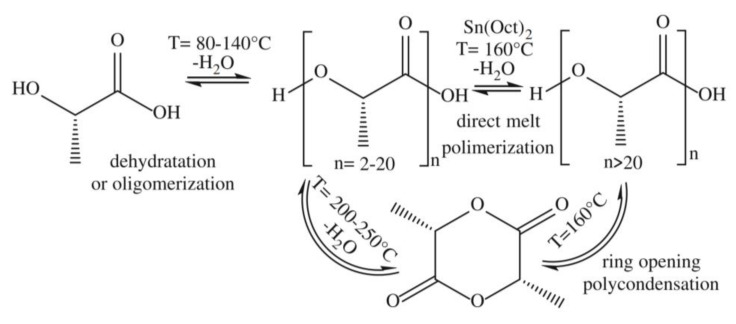
Synthetic path to produce Poly-(*l*-lactic acid) (PLA) from *l*-lactic acid [48].

**Figure 6 molecules-25-03829-f006:**
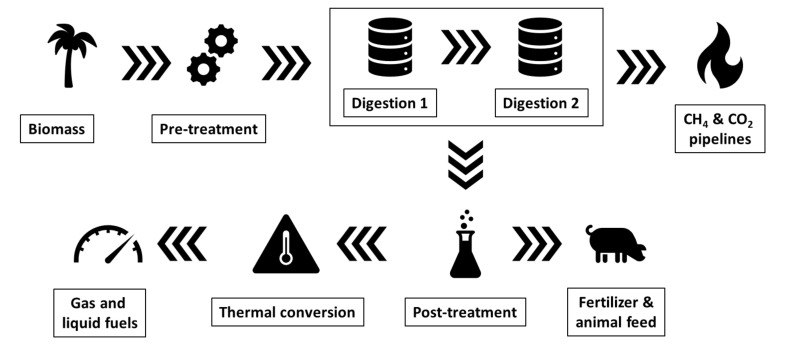
Schematics showing the process of banana waste anaerobic digestion to produce gas and liquid fuels, as well as fertilizer and animal feed (Modified from Figure 1 in Reference [15]).

**Figure 7 molecules-25-03829-f007:**
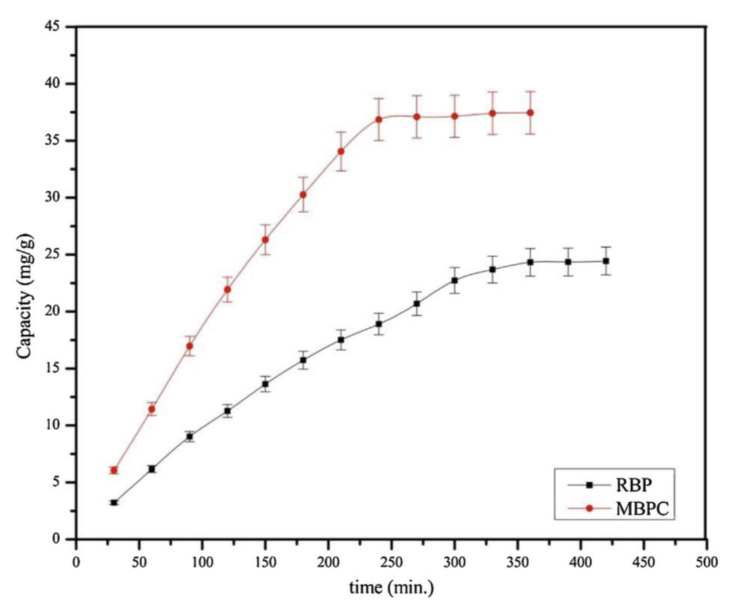
Removal efficiency of benzoic acid (BA) and salicylic acid (SA) from water samples using banana peel as adsorbent (C_0_ = 100 mg/L, t = 15 h, T = 303 K, modified from Reference [54]).

**Table 1 molecules-25-03829-t001:** Composition analysis of banana peel and rachis on a dry matter basis [20,23].

Parameter	Value (wt%) ^1^	
	Peel	Rachis
**Cellulose**	12.17 ± 0.21	23.0 ± 1.1
**Hemicellulose**	10.19 ± 0.12	11.2 ± 2.2
**(Acid-Detergent) Lignin**	2.88 ± 0.05	10.8 ± 0.5
**Sucrose**	15.58 ± 0.45	-
**Glucose**	7.45 ± 0.56	-
**Fructose**	6.2 ± 0.4	-
**Protein**	5.13 ± 0.14	-
**Pectin**	15.9 ± 0.3	-
**Ashes**	9.81 ± 0.42	29.9 ± 0.9

^1^ Polyphenolics, fat, and other extractives add up as the remainder of the composition.

**Table 2 molecules-25-03829-t002:** Estimated potential power generation based on whole banana residues and banana peel in Malaysia during 2003–2008.

Year	Whole Residue-Based Estimates ^a^				Peel-Based Estimates ^b^		
	Yield (kt/Year)	Energy from Biomass Residue (MJ/kg)	Energy Potential (PJ)	Potential Power Generation (MW)	Energy from Peel Residue (MJ/kg)	Energy Potential (PJ)	Potential Power Generation (MW)
2003	274	659	8.63	83.35	69	1.30	12.52
2004	317	761	9.97	96.31	79	1.50	14.47
2005	262	629	8.24	79.65	66	1.24	11.96
2006	258	620	8.13	78.50	65	1.22	11.79
2007	265	636	8.34	80.52	66	1.25	12.10
2008	270	649	8.5	82.13	68	1.28	12.34

^a^ Residue: Product Ratio = 2.4, ^b^ Peel: Product Ratio = 0.25, modified from Reference [15].

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
