# Peer review of "Biorefinery of Biomass of Agro-Industrial Banana Waste to Obtain High-Value Biopolymers"

_molecules, 2020, doi:10.3390/molecules25173829_

Round 1

Reviewer 1 Report

The manuscript presents an important and relevant review topic to readers of Molecules and more generally in the area of biological conversion of food waste.

This paper requires significant editing to capture the appropriate message and to be more precise. Language is important. For example, in the Abstract, garbage should be replaced with low value.

the paper needs to be better referenced, for example line 75-76 on severe wastes and line 55-58 on history of biorefineries, these passages require appropriate citations.

on the whole they paper presents an interesting story but needs extensive revisions.

Reviewer 2 Report

  • English editing would be beneficial to improve reader comprehension, particularly since the present paper is a review.
  • (Line 28-42) this entire paragraph has one single reference. Please include the adequate references, particularly for the definition of food supply chain waste
  • (line 50-51) the use of polymers in this sentence makes one thing of biopolymers, as in the subject of this review. Perhaps replace with polymeric substrates to underline that you are referring to the source material.
  • (Line 63-65) “Biorefinery can be considered as a facility, a process, a plant, or a cluster of facilities that 63 efficiently integrates the different biomass conversion technologies into biofuels, power (heat and 64 electricity), high-value products, and finest chemicals” as it is, this definition seems rather broad. A biorefinery aims to extract as much value as possible from a given source and such is not very clear here.
  • (line 75) Attention to the use of , and . to indicate millions and decimals.
  • (Line 83-87) This paragraph should be the introduction to the section as it introduces the why to the use of banana wastes far better. Please refer what is considered waste in the production of banana (leaves, peels, etc?)
  • (Line 88) Include the equivalent in mass for better understanding.
  • (Line 125-126) is this sentence supposed to mean lignin can be decomposed by anaerobic microorganisms?
  • (line 136) can biorefinery be called a technology? It is a concept to make biomass valorization viable and extract value from all streams in a biological process, not a technology per se.
  • (line 213-) please add some more of disadvantages of these types of energies since cost is not the only issue. Perhaps diversify references for this chapter.
  • (line 220-221) How is this gas produced? What do you mean with clean form of biogas? Low Sulphur concentrations?
  • Figure 4, if this graph is copied directly from the reference, be careful with copyright issues. If not and some alteration has been introduced, please change reference to (adapted from Ingale et al., 2014). Same issue should be addressed with figure 5.
  • (line 262) (50 and 500 mg/mL)
  • (Line 319) won’t the gasification of contaminated material worsen the quality of the gas and the waste streams of the gasification (ex: chars).
  • (line 334-339) please reformulate this section as to relate the 3 statements and expand on the PHB production process.
  • As a last addition, little has been underlined over the difficulty of using cellulosic and hemicellulosic materials, as well as the impact of some pretreatment processes upon the sustainability of the conversion into products of interest. This should be addressed.

Reviewer 3 Report

The paper scopes to overview the potential of banana waste to produce alternative energy sources and products with a high added value. The idea of the manuscript is beneficial, and I agree that banana waste has the potential to be treated in a biorefinery concept. Nevertheless, the paper is written very generally about biorefinery, wastes, products, with no more in-depth evaluation of banana waste potential. I have, therefore, these significant professional comments:

P1) First five of 15 pages are very general, and they are not associated with the topic. It is just public talk about wastes, biorefinery, waste to X potential. I agree with all the comments. Nevertheless, I think that authors should deal with a deeper overview of banana waste production and its characteristics (composition concerning locality, the effect of weather), and to discuss the profoundly current use of banana waste.

P2) Table 1 is ok. Nevertheless, total solids, moisture content and volatile solids should always be mentioned. Authors stated (line 103) that annual production was 2.9 million tons. Total solids or waste with natural moisture? All these values should be precise.

P3) Chapter 1.7.1 – Energy. The provided information about energy potential is based only on one reference. The discussion about biogas potential is served, no data are presented. Table 2 is generalized and irrelevant. More in-depth analysis is missing how energy potential was calculated, which process, efficiencies, technological set-up.

P4) Chapter 1.7.2. – Bioethanol – is also not overviewed in detail, just two references. Plenty of research was done, i.e. it could be fruitful to summarize waste, pre-treatment and conditions, alcoholic fermentation and bioethanol yield.

P5) I miss more profound technological overview in all subsequent chapters as discussed in P4.

P6) Relations banana waste, product yield, product content in waste, production cost, technological conditions, demand to a product, technical readiness level of maturity should be discussed in review papers.

Formal comments:

F1) Improve English. Some sentences are too long that it is hard to read and understand.

F2) Reference style in the text is not unified in the manuscript. Write it in one form.

F3) All the percentages have to be defined in the meaning, i.e. % wt., % vol., etc.

Reviewer 4 Report

This review article provides relatively comprehensive information on biorefinery of banana wastes. Before it can be accepted for publication, several comments are provided for consideration:

  1. The whole manuscript contains numerous short paragraphs with some paragraphs containing only one sentence. For example, section 1.3, 1.6, 1.7.3, 1.7.6. Some paragraphs that talk about similar topic should be combined together to make the manuscript better organized.
  2. The language use needs to be refined. Some inappropriate use of words makes understanding difficult. Examples are:

           Page 3, line 96-97, “to produce not only energy, but also biofuel production, …”

           Page 3, line 114, “WtW” needs to be defined somewhere.

           Page 4, line 121, “such as energy and chemical substances generation or biogas production”, this is confusing as biogas is a kind of energy and chemical substance.

           Page 5, line 171-173, “such as the production of biofuels, produce energy, water treatment, produce cosmetics and…”, should be re-organized.

           Page 7, line 199-201, “The banana stem also has …having the glucan and ash the highest percentage”, this sentence is confusing.

           Page 7, line 203-204, “… product you want to obtain…”, It is not recommended to use "you" in scientific writing.

  1. Some parts in the text are not matching what is shown in figures, examples are:

           Page 3, line 113, “and its blending with regular gasoline” is not indicated in Figure 1.

           Page 5, line 145, “banana residues are not found within…”, Banana residues should be in the "other" part in Figure 3. It would be more helpful if "other" part can be classified in a little bit detail, or the Figure 3 may not be relevant to the overall topic.

           Page 5, line 174, “animal feed” is not mentioned in Figure 4.

           Page 9, line 245, please note the figure number here, also please explain the unit of reducing sugar "mg%", please convert the unit of produced ethanol to concentration.

           Page 10, line 258, figure number. Lower case letter is in each sub-figure but it’s not properly explained in figure captions.

  1. Some sections of the whole manuscript are not organized well, examples are:

        Section 1.4, This whole part should be placed right after section 1.1 which talks about agroindustry residues.

        Section 1.7.1, the first two paragraphs are not related to the topic. For introduction of renewable energy, it should be placed in the beginning of the whole manuscript.

        Section 1.7.4, This section doesn't mention banana residues at all. What is the relationship between nanocellulose with banana residues?

        Section 1.7.5, Direct water treatment using banana waste cannot be considered as a biorefinery process.

Round 2

Reviewer 1 Report

The manuscript has improved but still would benefit greatly from English language editing to elevate the final composition of the paper. For example, the first sentence should be "On a worldwide scale...." rather than "In a ..." Subtle re-wording may not seem obvious except to native English readers, but editing the paper to improve even subtle aspects like this will make the paper accessible to a broader audience.

The paper has improved and I believe the new figures introduced have improved the paper.

The authors need to include their contributions as currently there is a generic statement from the template. 

Author Response

Comments:
The manuscript has improved but still would benefit greatly from English language editing to elevate the final composition of the paper. For example, the first sentence should be "On a worldwide scale...." rather than "In a ..." Subtle re-wording may not seem obvious except to native English readers, but editing the paper to improve even subtle aspects like this will make the paper accessible to a broader audience.

Author: We thank the reviewer for this observation. We have handed our manuscript to a native speaker and made significant improvements. All modifications are highlighted in green in the new document.

The paper has improved and I believe the new figures introduced have improved the paper.

Author: We thank the reviewer very much for their previous thorough revision and the pertinent comments.

The authors need to include their contributions as currently there is a generic statement from the template.

Author: This point has been addressed in our new version.

Reviewer 3 Report

The authors improved the manuscript and I am satisfied.

Author Response

Author: We thank the reviewer very much for their previous thorough revision and the pertinent comments.

Reviewer 4 Report

The manuscript has been significantly improved based on previous reviewers' comments. No further comments are provided for this revised manuscript.

Author Response

(The authors gave the same response as above.)
